# Identification of a guanine-specific pocket in the protein N of SARS-CoV-2

J. Rafael Ciges-Tomas [1,2✉], María Luisa Franco[1] & Marçal Vilar [1✉]

The SARS-CoV-2 nucleocapsid protein (N) is responsible for RNA binding. Here we report the crystal structure of the C-terminal domain ($N^{CTD}$) in open and closed conformations and in complex with guanine triphosphate, GTP. The crystal structure and biochemical studies reveal a specific interaction between the guanine, a nucleotide enriched in the packaging signals regions of coronaviruses, and a highly conserved tryptophan residue (W330). In addition, EMSA assays with SARS-CoV-2 derived RNA hairpin loops from a putative viral packaging sequence showed the preference interaction of the N-CTD to RNA oligonucleotides containing G and the loss of the specificity in the mutant W330A. Here we propose that this interaction may facilitate the viral assembly process. In summary, we have identified a specific guanine-binding pocket in the N protein that may be used to design viral assembly inhibitors.

[1] Instituto de Biomedicina de Valencia-CSIC Spanish National Research Council, C/Jaime Roig, 11, 46010 Valencia, Spain. [2] Present address: Structural Molecular Biology Group, Novo Nordisk Foundation Centre for Protein Research, Faculty of Health and Medical Sciences University of Copenhagen, Blegdamsvej 3-B, 2200 Copenhagen, Denmark. ✉email: rafaelciges@gmail.com; mvilar@ibv.csic.es

The severe acute respiratory syndrome coronavirus 2 (SARS-CoV-2) has emerged as a pandemic virus causing a global human health crisis. SARS-CoV-2 shares ≈80% genome identity with SARS-CoV[1]. At the 5′-terminus two-thirds of its genome, there are located the ORF1a and ORF1a/b encoding 16 non-structural proteins (NSPs 1-16), which are responsible for establishing the cellular conditions favorable for viral infection and viral mRNA synthesis[2,3]. The 3′-terminus one-third of the genome contains the ORFs for the four structural proteins: Spike (S), Envelope (E), Membrane (M), and Nucleo-capsid (N) and other accessory proteins[3]. S, E and M are trans-membrane proteins that are incorporated into the viral lipid envelope[4]. The protein N is a multifunctional protein that packs the genomic RNA and enhances the efficiency of virus tran-scription and assembly[5–13]. N contains an N-terminal domain, involved in RNA binding, and a C-terminal ($N^{CTD}$) involved in RNA binding, protein dimerization, and interaction with the viral membrane protein (M)[14,15]. In the case of the Murine Hepatitis coronavirus, MHV (a betacoronavirus) the $N^{CTD}$ has been implicated in the interaction with a conserved packaging signal (PS) necessary for viral RNA packaging[16].

To date, 11 crystallographic structures of the SARS-CoV-2 $N^{CTD}$ have been deposited at the protein databank presenting minor differences between them (PDB: 7N0I, 7F2B, 7F2E, 7C22, 7CE0, 6ZCO, 6YUN, 6WZO, 6WZQ, 6WJI, 7DE1, 7F2B, 7F2E). Some of them have been already published and others are pending publication[17–22], however, none of these contain RNA or nucleotides that provide any insight into the RNA recognition process. The SARS-CoV-2 $N^{CTD}$ folds into a helical core with a protruding β-hairpin employed for protein dimerization. In this study, we report three SARS-CoV-2 $N^{CTD}$ structures and provide a co-crystal structure with GTP that might provide an insight into the specificity of the PS recognition.

## Results

**Structure of SARS-CoV-2 $N^{CTD}$ in open and closed con-formations.** The crystal structure of $N^{CTD}$ was solved by molecular replacement to a resolution of 1.94 Å (Table 1). The crystal asym-metric unit contains two homodimers of $N^{CTD}$ with almost identical conformation (rmsd 0.14 Å). As described previously, it folds into five α-helices (α1–α5), two $3_{10}$ helices (η1–η2), and two β-strands (β1–β2), presenting the following N- to C-sequence for the struc-tural elements: η1–α1–α2–α3–α4–β1–β2–α5–η2[17–21] (Fig. 1a and Supplementary Fig. 1a). Two monomers are intertwined through the β-hairpin forming a four antiparallel β-sheet on one face of the dimer (Fig. 1). In striking contrast to the previously described structures, the superposition of the subunits in our structure shows a ≈5.5 Å movement of the β-hairpin (Fig. 1b). One subunit presents the β-hairpin in an extended and previously unseen conformation that we named "open". Whereas the other subunit shows a β-hairpin in a flexed conformation that we named "closed", similar to other structures of this protein (Fig. 1b and Supplementary Fig. 1b). The alternative conformation of the β-hairpin is associated with the structural displacement of the loop between α1 and α2 (residues 280–283) (Fig. 1b).

In the closed conformation the β-hairpin interacts with the C-terminal proline residue (P364) of a symmetry-related protein (Fig. 1c, d and Supplementary Fig. 1b). This interdimeric interaction between W330 and P364 through a π–π stacking (face-to-face rings interaction), and making hydrogen bonds with T325 and S327 (Fig. 1d), is a common feature with seven $N^{CTD}$ structures determined previously. But strikingly, our structure shows that in the open conformation, the side chain of the residue W330 is rotated 180° towards the β-hairpin making hydrophobic contacts with the side chain of the residues T325 and S327

(Fig. 1c). Therefore, our structure shows that upon a side-chain arrangement the β-hairpin changes from an open conformation to a closed conformation that might favor the interdimeric interaction.

Previous structures of SARS-CoV-2 $N^{CTD}$ showed that the β-hairpin that does not interact with the symmetry-related molecule has a molecule of acetate (PDB 7C22)[20] or sulfate (PDB 6WZQ)[19] interacting with the side chain of W330 (Supplementary Fig. 2a, b).

**Structure of SARS-CoV-2 $N^{CTD}$ bound to GTP.** Based on the structural data we hypothesized that W330 could be a suitable residue for RNA recognition, as W330 could interact with the phosphate moiety, similarly to the sulfate, or via π–π stacking with the nitrogen base. To test our hypothesis, we attempted to individually co-crystallize $N^{CTD}$ with the oxynucleotides UTP, ATP, CTP, or GTP. Although crystals were obtained in all the mixtures, only the GTP was co-crystallized with $N^{CTD}$. The structure of the binary complex $N^{CTD}$-GTP was solved in two different space groups, P2$_1$ and P1, to a resolution of 1.8 and 2 Å, respectively (Table 1). Both crystal forms contain two protein dimers in the asymmetric unit. However, in the space group P2$_1$ there is one molecule of GTP bound to one of the dimers, whereas in the P1 the asymmetric unit contains two molecules of GTP, each bound to one dimer (Fig. 2). The electron density map is well defined for the three GTP molecules, which present tem-perature factors that increase from the guanine moiety to the phosphates (Fig. 2 and Supplementary Fig. 3a), suggesting that the guanine is well anchored to the protein and the phosphates are more flexible. In fact, the phosphates β and γ in the crystal P1 show two alternative dispositions, one identical to the crystal P2$_1$ (Fig. 2). In the two crystal forms the GTP binds to a cleft between the two subunits of the dimer, and adjacent to the β-hairpin in the closed conformation (Fig. 3a). The cleft corre-sponds to a cavity adjacent and perpendicular to W330, reducing the accessibility of the tryptophan to the solvent (Fig. 3a and Supplementary Fig. 3b).

One side of the guanine ring establishes a π–π T-shaped interaction (edge-to-face) with the indole ring of W330 and hydrophobic interactions with K338 (Fig. 3a). The other side of the guanine ring stacks over the guanidinium group of R259, which makes hydrogen bonds with the OH– groups of the ribose ring (Fig. 3a). R259 and R262 are responsible for multiple hydrogen-bonding contacts with the β- and γ-phosphates (Fig. 3a). At the bottom of the cleft the guanine moiety interacts with the M317 and with the main chain of the residues K338 and A336 (Fig. 3a). These interactions suggest high specificity for guanine, as other nitrogen bases have not as many favorable interactions as the guanine (Supplementary Fig. 4). For instance, adenine lacks the C6 oxygen that makes interactions with the M317, K338, and A336 in the case of guanine (Supplementary Fig. 4). All protein–ligand contacts are summarized in the Supplementary Table 1. Importantly, the C-terminal tail of the symmetry-related dimer also contributes to the GTP binding, covering the cleft as a lid, with Van der Waals and hydrophobic interactions between T362′ and the ribose and with the π–π stacking between P364′ and W330 (Fig. 3a). The structure suggests that the GTP binding favors swapping of the β-hairpin and interdimeric interaction.

**Characterization of GTP binding.** To confirm our structural results, we studied the binding of GTP in solution using the intrinsic fluorescence of tryptophan. The protein contains two tryptophan residues: W330, located in the β-hairpin and exposed to the solvent, and W301 partially buried in the protein core

**Table 1 Data collection and crystallographic statistics.**

| SARS-CoV-2 | $N^{CTD}$ | $N^{CTD}$ in complex with GTP | |
|---|---|---|---|
| | | Crystal form I | Crystal form II |
| *Processed data* | | | |
| Beamline | XALOC (ALBA) | XALOC (ALBA) | XALOC (ALBA) |
| Wavelength (Å) | 0.97926 | 0.97926 | 0.97926 |
| Space group | P1 | $P2_1$ | P1 |
| *Cell dimensions* | | | |
| $a, b, c$ (Å) | 43.84, 44.82, 59.12 | 43.77, 92.72, 68.63 | 43.69, 48.41, 68.67 |
| $\alpha, \beta, \gamma$ (°) | 92.04, 96.20, 90.00 | 90.00 | 74.37, 89.89,83.17 |
| Resolution (Å) | 44.79–1.94 | 92.72–1.80 | 66.09–2.00 |
| | (2.04-1.94) | (1.84-1.80) | (2.05-2.00) |
| $R_{pim}$ (%) | 0.109 (0.527) | 0.027 (0.169) | 0.052 (0.212) |
| Mean $I/\delta(I)$ | 6.2 (2) | 17.8 (4.9) | 8.9 (3) |
| CC (1/2) | 0.984 (0.704) | 0.999 (0.961) | 0.993 (0.923) |
| Unique reflections | 32080 (4753) | 50472 (2926) | 34712 (2497) |
| Completeness (%) | 96.1 (96.6) | 99.4 (97.8) | 95.5 (93.2) |
| Redundancy | 3.4 (3.5) | 6.8 (6.4) | 3.4 (3.1) |
| *Refined data* | | | |
| $R_{factor}$ (%) | 0.185 | 0.165 | 0.192 |
| $R_{free}$ (%) | 0.239 | 0.211 | 0.245 |
| *RMSD* | | | |
| Bond deviation (Å) | 0.017 | 0.020 | 0.014 |
| Angle deviation (º) | 1. 828 | 1. 937 | 1. 748 |
| *Ramachandran map* | | | |
| Favoured (%) | 96.55 | 98.14 | 97.65 |
| Allowed (%) | 3.45 | 1.86 | 2.35 |
| Disallowed region (%) | 0 | 0 | 0 |
| *PDB accession code* | 7O05 | 7O35 | 7O36 |

Values in parentheses correspond to the data for the highest resolution shell.
$R_{pim} = \Sigma_{hkl}\sqrt{(1/(n-1))\Sigma_i|I(hkl)_i - \langle I(hkl)\rangle|}/\Sigma_{hkl}\Sigma_i\, I(hkl)_i$.
$R_{factor} = \Sigma||Fo| - |Fc||/\Sigma|Fo|$.
$R_{free}$ is the $R_{factor}$ calculated with 5% of the total unique reflections chosen randomly and omitted from refinement.

(Supplementary Fig. 5a). $N^{CTD}$ shows a fluorescence peak profile with a maximum fluorescence emission centered at 340 nm that decreases concomitantly with the increasing concentration of acrylamide, confirming the tryptophan fluorescence is quenchable (Fig. 3b and Supplementary Fig. 5b).

We also made the mutant $N^{CTD-W330A}$, replacing W330 with alanine, and confirmed that the fluorescence of the remaining W301 is also quenchable (Fig. 3b). We determined the quenching of the fluorescence in the absence or presence of 0.5 mM GTP. Whereas GTP decreased significantly the quenching slope of $N^{CTD}$ (from $6.31 \pm 0.08$ to $5.17 \pm 0.09$; p-val ****, $n = 7$ and $n = 4$), the quenching of $N^{CTD-W330A}$ ($1.99 \pm 0.06$; $n = 4$) is not affected by the presence of GTP ($1.74 \pm 0.06$; $n = 4$), indicating that only the accessibility of W330 is affected by GTP (Fig. 3b and Supplementary Tables 2 and 3). This effect is not observed in the presence of ATP or UTP, but it is shown in the presence of CTP to a much lesser degree (p-val **) (Supplementary Fig. 6 and Supplementary Table 3). We hypothesized that the binding of GTP to the homodimerization interface could stabilize the protein dimer. To confirm this hypothesis, we perform differential scanning fluorimetry (DSF) experiments with $N^{CTD}$ and $N^{CTD-W330A}$ in the presence or absence of 5 mM GTP (Fig. 3c). $N^{CTD}$ presents a sigmoidal trajectory in response to temperature, with a melting temperature ($T_m$) of 48.96 °C (±0.03) that increased to 49.97 °C (±0.03) in presence of GTP (Fig. 3c). This difference is statistically significant (Aikake test, see the "Methods" section). In contrast, the mutant $N^{CTD-W330A}$ shows a $T_m = 47.98$ °C (±0.07) unaffected by the presence of GTP, $T_m = 47.67$ (±0.07) (Fig. 3c).

We also measure the binding affinity ($K_d$) for GTP of $N^{CTD}$ and $N^{CTD-W330A}$ using Microscale Thermophoresis (MST)

(Fig. 3d and Supplementary Table 4). $N^{CTD}$ presents a $K_d$ value of 196 μM for GTP, which increases to 858 μM in the mutant $N^{CTD-W330A}$. This result confirms the participation of W330 for the GTP binding in solution (Fig. 3d). Although the affinity for GTP is in μM range, it should be taken into account that this $K_d$ value corresponds to the affinity of one nucleotide of a protein that binds RNA, in which a synergistic effect of multiple nucleotide interactions would favor the avidity (see below for RNA hairpin binding affinity).

**W330 confers specificity of the N-CTD to the guanine-containing RNA oligonucleotides.** Phylogenetic analysis suggests the existence of conserved short structured repeats, termed repetitive structural motifs or RSM in the gRNAs from coronaviruses, that may facilitate the genome packaging by specific interaction with protein factors[23]. These RNA elements are called stem-loop (SL) 1–6 and SL5 is the most interesting as displays a conserved sequence 5′-UUYC**G**U-3′ in the tripartite apical substructures, SL5a–c. Interestingly these sequences contain a highly conserved G (in bold) in nearly all alpha- and beta-coronavirus[23].

We analyzed the binding of the N-CTD to an RNA oligonucleotide derived from the apical region of the stem-loop 5a (SL5a) of SARS-CoV-2 gRNA (Oligo-G, Fig. 4). Incubation of increasing concentrations of $N^{CTD}$ to the Oligo-G showed the binding and the protein–RNA complex formation, $K_d = 32 \pm 5$ nM, (Fig. 4). Interestingly mutation of the only G to U, Oligo-U, induces a decrease in the binding affinity, $K_d = 140 \pm 29$ nM (Fig. 4a, b). To study the role of the W330, we generated and purified the mutant $N^{CTD-W330A}$. EMSA assays showed that the mutant W330A binds less efficiently to the oligonucleotide

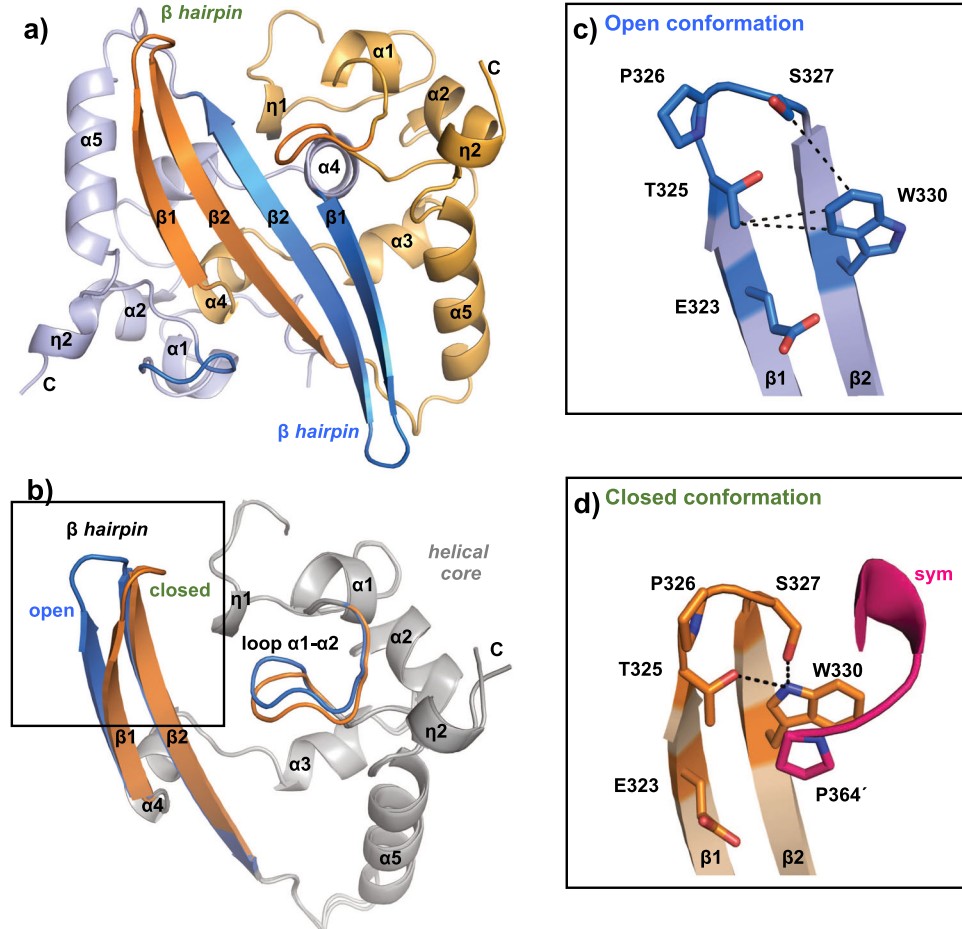

**Fig. 1 Crystal structure of SARS-CoV-2 NCTD in open and closed conformations. a** Cartoon representation of the N$^{CTD}$ dimer. Each monomer is colored in blue and orange, respectively. The β-hairpin and the loop connecting the helices α1–α2 are highlighted in dark tones. Secondary structural elements and residues are numbered and labeled in order from N to C terminus: the symbol η corresponds to 3$_{10}$ helix; α to α-helix and β to β-strand. **b** Superimposition of the N$^{CTD}$ monomers. The common helical core is colored in gray. **c, d** Detailed view of the open (**c**) and the closed (**d**) conformations. The side chain of key residues is shown in sticks with carbon atoms colored according to the monomer to which they belong. Hydrophobic and polar interactions are represented as dashed black lines. The C-terminal of the symmetric molecule (*sym*) is colored in magenta. The symmetric Pro residue is indicated with an apostrophe. Nitrogen and oxygen atoms are colored in blue and red, respectively.

Oligo-G ($K_d = 130 \pm 1$ nM, $n = 3$) than the NCTD-wt to Oligo-G ($K_d = 32 \pm 5$ nM, $n = 3$), and with a similar affinity to Oligo-U (W330A:Oligo-U $K_d = 90 \pm 19$ nM, $n = 3$ and NCTD-wt:Oligo-U $K_d = 140 \pm 34$ nM, $n = 3$), (Fig. 4a and b). This supports the specific recognition of the guanine in the RNA hairpin by the N-CTD and that the mutation W330A losses this specific recognition.

## Discussion
In coronavirus assembly of the genomic RNA, gRNA, is a fundamental problem as infection produces several copies of subgenomics RNAs, sgRNA. The specific interaction of N to PS sequences will assure the packaging of gRNA versus other sgRNAs, mRNAs from the host and other RNAs from the infected cells[24]. In the alpha- and beta-coronavirus, like SARS-CoV, SARS-CoV-2 and MERS-CoV phylogenetic data suggest the conservation of several of the RNA stem-loops (SL) such as SL5a-c that might constitute regions that participate as authentic PS[23]. Although we did not succeed to co-crystallize N$^{-CTD}$ with the SL5a hairpin, we found that the N$^{CTD}$ is able to specifically recognize the SL5a apical sequence from SARS-CoV-2, supporting previous suggestions indicating that this protein domain might be the region of the N protein able to recognize specific

features of coronavirus gRNA[24]. If the C-terminal domain of the M membrane protein of SARS-CoV-2 is implicated in this recognition as others have found in murine coronavirus[16,24], will need further studies.

Comparison to other human CoVs N$^{CTD}$ reveals that there are fifteen invariant residues among human coronaviruses, three of which belong to the identified GTP binding pocket (R259, R262, F274; numbers corresponding to SARS-CoV-2) (Fig. 5). The pocket is conserved in SARS-CoV-1, and presents only one conservative change in MERS that intriguingly correspond to the swinging W330, which changes to F (Fig. 5). The structural comparison with SARS-CoV-1 (PDB 2GIB)[25], MERS (PDB 6G13)[26], and NL63 (PDB 5EPW)[27] further suggests that the cleft is also conserved in other human coronaviruses (Fig. 5a–d). Intriguingly, the equivalent swinging tryptophan in SARS-CoV-1 also presents two rotamers, suggesting that the conformational mechanism of the β-hairpin described here could be common for both viruses (Fig. 5a, b). Moreover, the structure of MERS N$^{CTD}$ shows a molecule of trimethylamine oxide (TMO) that mimics the interactions shown here with the phosphates of GTP, strongly suggesting that the cleft of MERS N$^{CTD}$ could also be a suitable pocket for ligand binding (Fig. 5c).

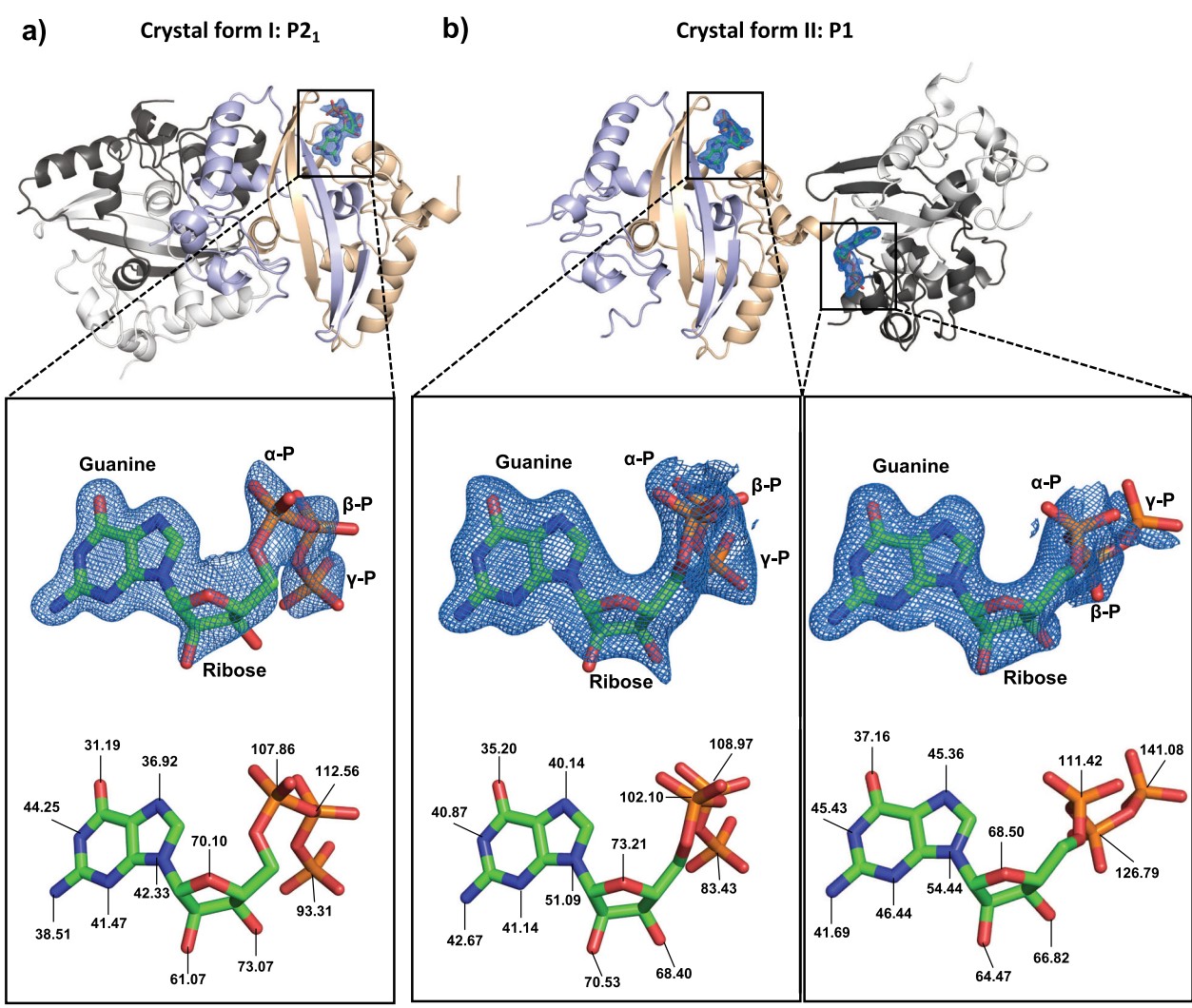

**Fig. 2 Structures of SARS-CoV-2 NCTD in complex with GTP.** Cartoon representation of the two dimers contained in the asymmetric unit of the crystal in the space group P2$_1$ (**a**) and P1 (**b**). Each monomer is colored in white and black for one dimer, and orange and blue for the second dimer. GTP is represented in sticks with carbon atoms colored green. The electron density map 2Fo−Fc ($\sigma = 1$) of the GTP is represented in blue. The moieties guanine, ribose, and phosphates are labeled. Nitrogen, oxygen, and phosphorus atoms are colored in blue, red, and orange, respectively. The temperature factors ($B$-factors) of nitrogen and oxygen atoms of the nucleoside and phosphorus atoms of phosphates are indicated below.

Moreover, our results suggest that guanine binding would promote the oligomerization of the N protein through interdimeric interactions to form the ribonucleoparticle. This sequence and structural conservation expand the possibility of using this guanine-binding pocket as a therapeutic target against SARS-CoV-2, SARS-CoV-1, highly probable against MERS, and possibly against previous human coronaviruses as NL63. In addition, the comparison of the guanine-binding pocket of N$^{CTD}$ with structurally similar human proteins that also bind guanine nucleotides, such as Rho6, RaI-A, and CD38 (Supplementary Fig. 7) shows low conservation in the set of residues and their structural organization for the nucleotide binding, suggesting that the design of specific compounds against SARS-CoV-2 N$^{CTD}$ may be feasible.

## Methods

**DNA methods**. The synthetic gene for the SARS-CoV-2 N$^{CTD}$ (residues 256–364) cloned into the vector pET28a(+) was provided by Biomatik. The insert is in frame with the N-terminal His$_6$-tag, and a TEV protease cleavage site. This construction was used as a template to introduce the mutation W330A using the Q5-Site Direct Mutagenesis Kit (NEB) with the oligonucleotides W330A-fw 5′-GAGCGG-TACCGCCCTGACCTATACCG- and W330A-rv 5′-GGGGTAACTTC-CATGCCA-. The final construct was sequenced in the IBV-Core Sequencing service.

**Protein production and purification**. *E. coli* BL21 (DE3) (Novagen) cells transformed with N$^{CTD}$ and N$^{CTD-W330A}$ constructs were grown at 37 °C in LB medium supplemented with 33 g/ml kanamycin to exponential phase (optical density of O.D. = 0.6, measured at $\lambda = 600$ nm), and then, protein overexpression was induced by addition of 1 mM isopropyl-β-D-1-thiogalactopyranoside (IPTG) for 16 h at 20 °C. After induction, cells were harvested by centrifugation at 4 °C for 30 min at 4000 × $g$, resuspended in buffer A (150 mM Tris–HCl pH 7.5, 250 mM NaCl) supplemented with 1 mM phenylmethanesulfonyl fluoride (PMSF) and lysed by sonication. The clarified lysate was loaded onto a 5 mL His Trap HP column (GE Healthcare) pre-equilibrated in buffer A. Following through wash with 20 column volumes of buffer A supplemented with 10 mM imidazole, the protein was eluted with buffer A supplemented with 500 mM imidazole. The sample was concentrated using an Amicon Ultracentrifugation device with 10 kDa cutoff, and dialyzed against buffer B (25 mM Tris–HCl pH 7.5 and 250 mM NaCl). The dialyzed protein was flash-frozen in liquid nitrogen and stored at −80 °C.

**Fluorescence spectroscopy measurements**. Protein fluorescence quenching was determined in a TECAN Spark instrument using acrylamide as a quencher. Samples were prepared at 5 μM protein and increasing concentration of acrylamide (0–1 M) in the absence or presence of 0.5 mM GTP. Samples of protein with or without GTP, and the acrylamide were prepared by separate at 2× in buffer B. Equal volumes of sample and acrylamide were mixed to a final volume of 50 μl (1×). Sample fluorescence was measured in 96-well black plates exciting at $\lambda_{ex} = 295$ nm (bandwidth of 5 nm; step size of 1 nm) and collecting the emission between $\lambda_{em} = 312$ and 380 nm (bandwidth of 5 nm; step size of 1 nm).

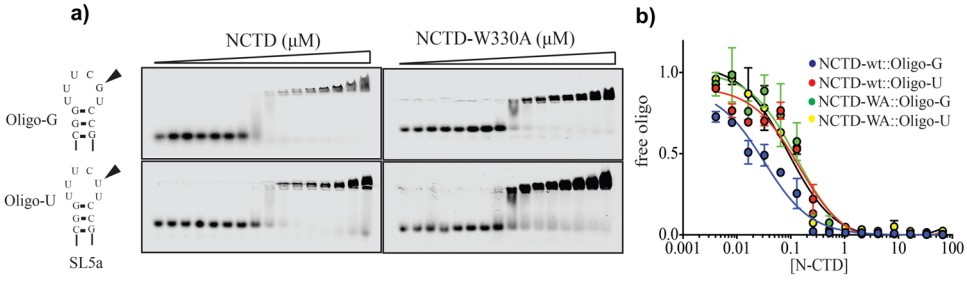

**Fig. 3 Crystal structure of SARS-CoV-2 NCTD in complex with GTP. a** Upper panel, the $N^{CTD}$ dimer in complex with GTP is represented in the cartoon. Each monomer is colored in blue and orange, respectively. The β-hairpins of the dimer are labeled and highlighted in dark tones. The GTP molecule is shown in sticks with its electron density map 2Fo−Fc ($\sigma = 1$) in green color. The close view of the GTP binding site is represented below. The side chain of key residues and the GTP molecule are shown in sticks with carbon atoms colored according to the monomer to which they belong. H-bond interactions of the ligand are represented as dashed black lines. The C-terminal of the symmetric molecule (*sym*) is colored in magenta with residues indicated with apostrophes. Secondary structural elements are numbered and labeled in order from N to C terminus. Nitrogen, oxygen, and phosphorus atoms are colored in blue, red, and orange, respectively. **b** Quenching of fluorescence (Fo/F) represented against increasing concentration of acrylamide for the $N^{CTD}$ and $N^{CTD\text{-}W330A}$ proteins in the presence or absence of GTP. Trajectories are fitted to linear regression. Statistical differences are indicated with asterisks (****$p < 0.0001$, n.s. $p > 0.05$). **c** Thermal unfolding curves of $N^{CTD}$ and $N^{CTD\text{-}W330A}$ in the presence or absence of GTP. Trajectories are fitted to a sigmoid. The corresponding melting temperature ($T_m$) is indicated. **d** GTP binding quantified thermophoretically. GTP is titrated to a constant amount of fluorescently labeled $N^{CTD}$ and $N^{CTD\text{-}W330A}$. The binding affinity ($K_d$) is indicated. The error bar represents the standard error of the mean of at least four experiments.

**Fig. 4 Binding of NCTD to the apical region of SL5a RNA. a** Ag-EMSA gels of titration experiments. Binding of $N^{CTD}$ and $N^{CTD\text{-}W330A}$ to an ssRNA oligonucleotide from the apical part of the SL5a region of SARS-CoV-2 gRNA with conserved guanine (arrowhead, Oligo-G) or with a G to U mutation (Oligo-U, lower panels). **b** ssRNA oligonucleotide shift with increasing concentration of protein. Protein–ssRNA combinations are represented in different colors. Trajectories are fitted to a sigmoidal one-site binding model. The error bar represents the standard error of the mean of at least four experiments.

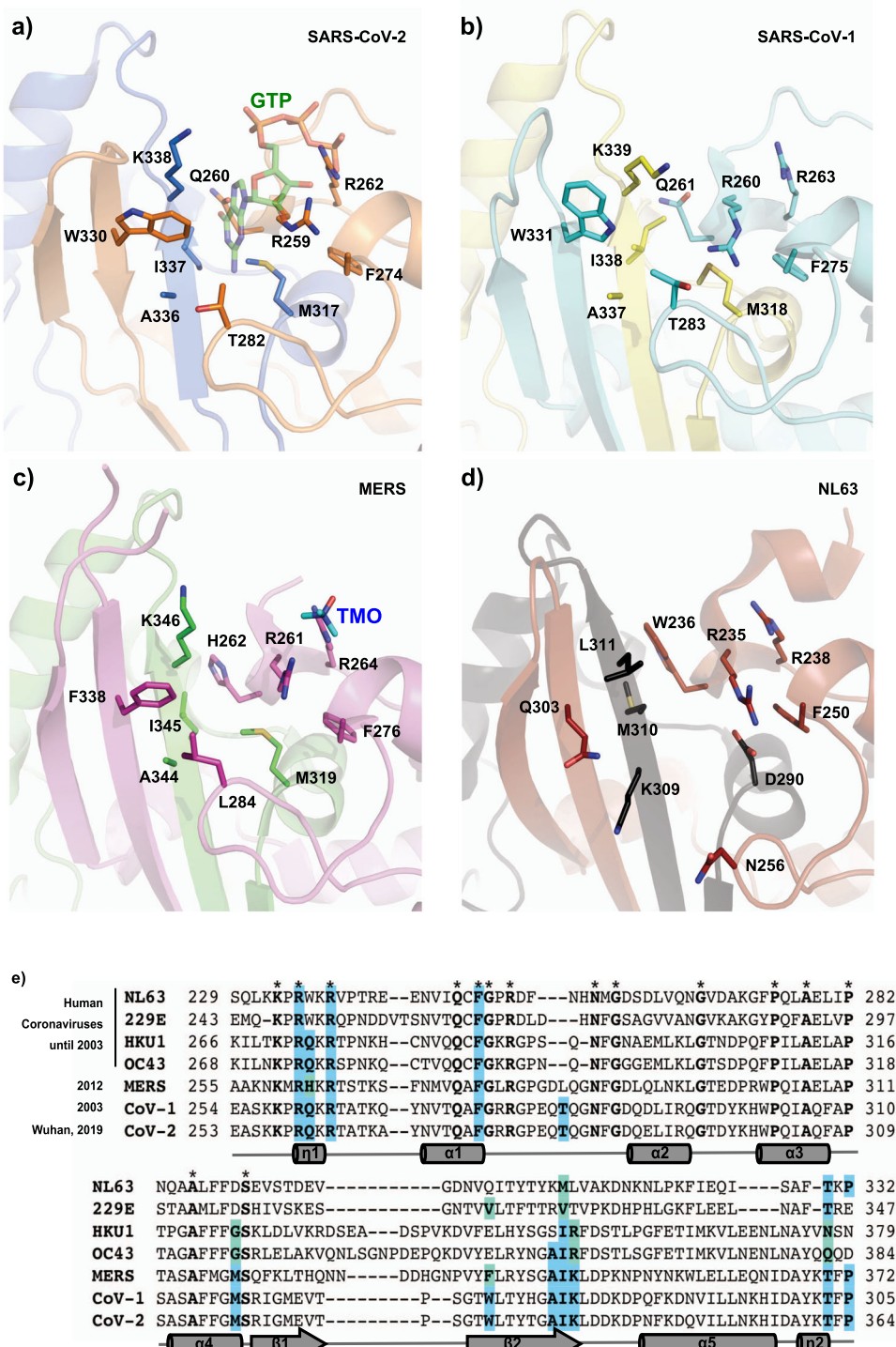

**Fig. 5 Conservation of the Guanine binding pocket in human coronaviruses.** Cartoon representation of the GTP binding pocket in SARS-CoV-2 (**a**), SARS-CoV-1 (PDB 2GIB) (**b**), MERS (PDB 6G13) (**c**) and NL63 (PDB 5EPW) (**d**). Each monomer of the dimer is represented in different colors. The residues that mediate the GTP binding in SARS-CoV-2 and equivalent residues in other coronaviruses are shown in sticks with carbons in the same color as the monomer to which they belong. The GTP molecule is represented in semi-transparent sticks with carbon atoms in green. The trimethylamine oxide (TMO) bound to MERS is shown in sticks with carbons in cyan. **e** Sequence alignment of N$^{CTD}$ of human coronaviruses. Invariant residues are in bold and indicated with an asterisk above the sequence. Residues for the GTP binding are in bold and highlighted in blue, and conservative changes are in light green. Secondary structural elements of the SARS-CoV-2 N$^{CTD}$ are represented below the sequence and labeled and numbered from N to C. η corresponds to 3$_{10}$ helices; α to α-helices and β to β-strands.

The emission at 340 nm was used for calculation. Quenching was calculated by dividing the fluorescence of the sample in absence of acrylamide (Fo) by the fluorescence of the sample in the presence of acrylamide (F), (ratio Fo/F) at each acrylamide concentration.

**Differential scanning fluorimetry (DSF)**. The structural integrity of the protein at increasing temperature was performed using SYPRO Orange as an external fluorophore in a 7500 Real-Time PCR System (Applied Biosystems). 20 μl samples were prepared at 5x SYPRO Orange fluorophore (Sigma), 80 μM of protein and 5 mM GTP (when indicated) in buffer B. Samples were loaded in 96-well PCR plates and heated from 20 to 85 °C with a heating rate of 1 °C/min. Fluorescence intensity was normalized and analyzed using GraphPad Prism software.

**MicroScale termophoresis (MST)**. His-tagged N$^{CTD}$ was fluorescently labeled with NT-647 amine-reactive dye from the Monolith NT Protein Labeling kit RED-NHS (NanoTemper Technologies) according to the manufacturer's instructions. The protein was prepared at 20 μM in 250 mM NaCl and mixed with 3x dye solution (1:1 volume), the mixture was incubated 30 min in dark at room temperature after soon the labeled protein was purified to remove free dye by gravity column with buffer C (25 mM Tris–HCl pH 7.5, 250 mM NaCl and 0.1% pluronic acid F127). Labeled N$^{CTD}$ protein was used at a final concentration of 20 nM. A 16-point twofold dilution series (ranged from 50 mM) of GTP in buffer C was mixed with N$^{CTD}$-labeled solution (1:1), and incubated 15 min at room temperature. Samples were filled into premium-coated capillaries (K005, NanoTemper Technologies) and MST measurement was performed on a Monolith NT.115 (Nano-Temper Technologies) at 25 °C in RED channel using a 77% LED excitation power and MST power of 40%. Analyses were performed with M.O. Affinity Analysis software (NanoTermper Technologies)[28] using the signal from an MST on-time of 1.5 s.

**Crystallization**. Crystals were grown in sitting drops at 21 °C with a vapor-diffusion approach. Crystallization trials were set up in the Crystallogenesis service of the IBV-CSIC using commercial screenings JBS I and II (JENA Biosciences) and JCSG+ (Molecular Dimensions) in 96-well plates. Crystallization drops were generated by mixing equal volumes (0.3 μl) of each protein solution and the corresponding reservoir solution and were equilibrated against 100 μl reservoir solution. His tagged N$^{CTD}$ was crystallized at a protein concentration of 8 mg/ml in a reservoir solution consisting of 0.1 M tri-sodium citrate pH 4.5; 0.1 M bisTris pH 5.5 and 25% PEG 3350. Crystals were cryoprotected with a solution consisting on reservoir solution supplemented with 10% glycerol.

For crystallization in complex with GTP, his tagged N$^{CTD}$ at 15 mg/ml was incubated overnight at 4 °C with 5 mM GTP. Crystals were grown in a reservoir solution consisting on 30% PEG 3350 and 0.1 M sodium acetate pH 4.6. Crystals were cryoprotected with a solution consisting of 30% PEG 3350, 10% glycerol, and 0.1 M sodium acetate pH 4.6. Both crystal forms (P2$_1$, P1) were obtained within the same crystallization drops.

**Data collection and model building**. X-ray diffraction was performed at 100 K in an XALOC beamline at ALBA synchrotron (Barcelona, Spain). Processing of collected data collection was performed with XDS and iMosflm programs[29,30], and the data were reduced with Aimless[31] from the CCP4 suite[32]. The data-collection statistics for the best data sets used in structure determination are shown in Extended Data Table 1. In crystal form I, the spacegroup P2$_1$ was selected during automatic data processing, and although orthorhombic space groups were also explored for this data set, no correct solutions were found in that crystal system, likely due to the fact that the two dimers contained in the asymmetric unit are not entirely identical and no higher symmetry can be applied. Phases were obtained by molecular replacement using Phaser program[33] and PDB 6WZO[19] as search model. Model building and refinement were done by iterative cycles using Refmac[34] and Coot[35]. Data refinement statistics are given in Extended Data Table 1.

**Electrophoretic mobility shift assay (EMSA)**. Single strand oligonucleotides 5'-CGGUUUC**G**CCG-3' and 5'-CGGUUUC**U**CCG-3' labeled on 5' with IRDye-800CW fluorophore (IDT) were used at a final concentration of 300 nM and serial dilution of protein ½ from 95 to 0 μM protein. Mixtures were prepared at a final volume of 10 μl and incubated at room temperature for 20 min. EMSA was performed on agarose 0.5% gels in acidic conditions on a buffer consisting of 0.355% (m/v) β-alanine adjusted to pH 4 with glacial acetic acid. 3 μl of methyl green loading buffer (50% glycerol, traces of methyl-green, 0.15% (v/v) glacial acetic acid adjusted to pH 5 with KOH) were added to the mixtures and gels were run at 4 °C for 20 min at 100 V. Gels were revealed in an Odyssey Imaging System (LI-COR) and images analyzed using ImageStudio. Quantification was done by normalizing the signal from the free oligo to 1.0 at 0 μM protein and plotting the free oligo versus the concentration of the N-CTD protein (in Log scale). Data were adjusted to a nonlinear curve of the form One site equation (Prism 6 software). Data of at least three independent experiments were plotted. Error bars represent the standard error of the mean.

**Statistics and reproducibility**. Statistical analyses were used in all graphs represented in Figs. 3 and 4. Figure 3b was analyzed using a one-way ANOVA with Tukey's multiple comparison test. Figure 3c was analyzed by comparing the nonlinear fitting curve using the Aikake test (AIC) with a >99.9% probability that the $T_m$ of N and $T_m$ of N + GTP is different, with an AIC of 221.1 (using GraphPad prism 6.0). In Fig. 4 statistics was analyzed by comparing the nonlinear fitting curve (one site equation in GraphPad Prism 6.0) using the Aikake test (AIC) with a >99.9% probability that the $K_d$ of the binding of N-CTD to Oligo-G is statistically different from the $K_d$ of the binding of N-CTD to Oligo-U is with an AIC of 24.1.

## Data availability

The coordinates and structure factors of the crystallographic structures have been deposited in the Protein Data Bank with accession codes PDB 7O05, 7O35 and 7O36. Graph raw data and gel images are deposited at Mendeley Data at the address https://data.mendeley.com/datasets/8cfsrmd8by/draft?a = a0d0ad46-dfb5-474e-b9c5-37c59410e6a3.

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

## Acknowledgements

We would like to thank to Dr. Encarnación Martínez-Salas from CBM-CSIC for her suggestions on the EMSA assays, and Dr. Santiago Ramón-Maiques and Dr. Francisco Del Caño-Ochoa from IBV-CSIC, for their technical support and critical reading of the manuscript and the staff of the synchrotron for technical assistance. The X-ray diffraction experiments were performed in XALOC beamline at ALBA Synchrotron (Barcelona; Spain). This work was supported by the COVID research grant COV20/01265 awarded to M.V. from the Instituto de Salud Carlos III (ISCIII; Spain) and by the European Commission–NextGenerationEU (Regulation EU 2020/2094), through CSIC's Global Health Platform (PTI Salud Global). X-ray diffraction data collection was supported by the Spanish Synchrotron Radiation Facility ALBA through the COVID Proposal 2020074407 awarded to J.R.C.-T. and M.V.

## Author contributions

Conceptualization: J.R.C.-T. and M.V.; Methodology: J.R.C.-T., M.L.F., and M.V.; Investigation: J.R.C.-T. and M.V.; writing-original draft, J.R.C.-T. and M.V.; funding acquisition: M.V.

## Competing interests

The authors declare no competing interests.
