## [Peer Review File · Communications Biology]

Reviewers' comments:

Reviewer #1 (Remarks to the Author):

This paper from Ciges-Tomas and colleagues presents structures of the N-protein from SARS-CoV-2 in apo and GTP liganded forms. They identify a potential GTP binding site for this protein which may play a role in viral assembly. The paper in general is well written, the work of interest, and the experiments mostly sound. I have a few queries and suggestions for the authors that I detail below.

Line 83 – References for the 7 structures previously determined should be included here.

Figure 3a. It is very unusual for 2fo-Fc maps to be colored green, green is usually used for positive density in difference density maps (eg Fo-Fc). The color should be changed to avoid confusion.

Also, ideally, simulated anneal omit maps should also be provided of the ligand to present maps free of any bias.

Figure 3b (and other similar experiments). Are these error bars and statistical evaluation based on multiple independent experiments (ie, not technical replicates). If they are then it should be clearly stated and the number of independent repeats described in the legend. If not then it should be repeated and significance evaluated across independent experiments.

Table 1 – only one angle is given for the P21 space group of N-CTD in complex with GTP. I am guessing the authors are referring to the Beta angle being 90, but should be clear on this. If the Beta angle is exactly 90.00 it is unusual, did the authors explore possible orthorhombic spacegroups? Some discussion on the determination of spacegroup would be useful in the methods or result section for clarity because of this.

Table 1 - for the N-CTD + GTP crystal form 1, potential data has not been included in the processing, data is truncated at an I/SigI of 4.9 in the outer shell. Is there a reason for this?

Minor Grammatical suggestions

Line 55 – consider changing to ",however, none of these contain RNA or ..."

Line 87 – check grammar on "hairpin change from an open conformation to a closed conformation .."
consider changing to "hairpin changes from .."

Line 183 – check grammar for "and that the mutation W330A loss this specific recognition."

Reviewer #2 (Remarks to the Author):

The manuscript described the molecular basis of GTP binding to SARS-CoV-2 N protein using protein chemistry, mutagenesis, binding affinity, and structural biology. Overall the work is clear, conclusion supported by the data and the methods are well described. The findings are interesting as this represent the first structure of the N protein form SARS-CoV-2 binding GTP and describing its specificity over other nucleotide. This could help design some drug to block the binding and disrupting the RNA packing that is the principal function of the N protein.

Comments

Table I: the beta angle for the crystal form I of NCTD crystallised in the space group P21 should be

different from 90degree, there is probably a typo here. Also it look like this data set was cut in resolution with different rules than the two other data set and could have been pushed further. Can the authors please clarify?

Page4: the authors described 2 alternate conformation of the beta hairpin, with the closed conformation forming interaction with the symmetrical in the crystal packing, would this conformation be due to crystal packing? Or is only stabilised by the crystal packing?

Figure 2. The second GTP in the crystal form II (P1) seems to show weak density for the last PO4 group. Given this how clear was the omit maps for each GTP? Are the B factor of R259 and R262 similar to the PO4-gamma of the GTP? If not why?

Did the protein dimerise in solution? Or during crystallisation?

Line152-153 does the 1 degree difference in Tm significant?

In addition to the SL5a does the RNA of coronaviruses rich in G?

If drugs were to be developed against this GTP binding, how different is the GTP binding of the NCTD compare to other GTP binding protein (specially human one)?

Reviewer #3 (Remarks to the Author):

The manuscript "The identification of a guanine-specific hydrophobic pocket 1 in the N protein from SARS-CoV-2" by Ciges-Tomas, et al. reports crystal structure of the C-terminal domain of SARS-CoV-2 nucleocapsid (N) protein, N-CTD, and the protein in complex with GTP. They identify W330 as a crucial residue for GTP interaction and W330A mutant abrogated the binding of oligonucleotides. Overall, the work is of good quality. However, the language needs improvement. My specific comments are:

1. On page 6, line 122: "... and other nitrogen base would not be easily accommodated in the pocket." The authors should show in Supplementary Figure that there will be steric hindrance when other bases (A, U or C) are put in that binding site.

2. Page 6, line 123-128: ".....The structure suggests that the GTP binding favours swapping of the β -hairpin and interdimeric interaction." What do authors mean by swapping of beta-hairpin and inter dimeric interactions? The C-terminal tail of a symmetry-related dimer coming close to GTP binding site would be a result of crystal contacts and would not enhance the binding of GTP.

3. Page 7, line 142: "Whereas GTP decreased significantly the quenching slope of NCTD (from 6.31 ± 0.08 to 5.17 ± 0.09 ; p-val ****)" The authors mention the p-value but number of replicates for the experiments should also mentioned either in the main test or in Supplementary information.

4. Page 8, line 179-182: "EMSA assays showed that the mutant binds less efficiently to the oligonucleotide Oligo-G than the NCTD-wt ($K_d=130 \pm 34$ nM, n=3), and with a similar affinity to Oligo-U ($K_d=90 \pm 19$ nM)". It appears that the K_d for oligo-G binding to Nctd-wt is 130 nM, which is not true. The sentence may be reframed....

RESPONSE TO THE REVIEWERS

First of all, we are very grateful to the Editor and the three reviewers for their expertise and opinion. Below we provide a point-by-point response to all of the comments.

Reviewer #1 (Remarks to the Author):

This paper from Ciges-Tomas and colleagues presents structures of the N-protein from SARS-CoV-2 in apo and GTP liganded forms. They identify a potential GTP binding site for this protein which may play a role in viral assembly. The paper in general is well written, the work of interest, and the experiments mostly sound. I have a few queries and suggestions for the authors that I detail below.

Line 83 – References for the 7 structures previously determined should be included here.

We thank the reviewer for the suggestion. We have included the references as suggested. We also added references to 3 more structures of SARS-Cov-2 N-CTD that were recently released. All the PDB entries are cited in the text, although some of them are not described in an accompanying publication.

Figure 3a. It is very unusual for $2F_0$ -Fc maps to be colored green, green is usually used for positive density in difference density maps (eg F_o -Fc). The color should be changed to avoid confusion.

We thank the reviewer for the advice. The $2F_0$ -Fc map in Figure 3a has been changed to blue color. The new Figure 3a is shown here:

Also, ideally, simulated anneal omit maps should also be provided of the ligand to present maps free of any bias.

Thanks for the commentary. We agree with the reviewer that omit maps are more indicative of the presence of the ligand and are not biased by the model. Following the advice, we calculated

Asymmetric unit content of the crystal form I (s.g.P2₁)

Additionally, we have added the following explanation to the section “Data collection and Model Building” in Material and Methods:

“In the crystal form I, the spacegroup P2₁ was selected during automatic data processing, and although orthorhombic space groups were also explored for this data set, no correct solutions were found in that crystal system, likely due to the fact that the two dimers within the asymmetric unit are not identical.”

Table 1 - for the N-CTD + GTP crystal form 1, potential data has not been included in the processing, data is truncated at an I/SigI of 4.9 in the outer shell. Is there a reason for this?

We thank the reviewer for this comment and we agree that according to the Mean I/SigI, the data can be truncated to a higher resolution and potential data could be included. When data were truncated at Mean I/sigI=1.1, corresponding to 1.56 Å resolution (picture included below), the completeness of the data in that high-resolution cell is 44%, and other statistical values are worse, such as Rpim= 0.58, and CC1/2 = 0.57. Considering the statistical data altogether, in our opinion the resolution 1.56 Å is a little bit optimistic, and the data were truncated to 1.8 Å resolution, at which the highest resolution cell has a completeness= 97.8% and CC1/2= 0.96, and at the same time ensures high quality values for other statistics such as Mean I/sigI and Rpim. Thereby, all three data sets have a completeness >90%, Mean I/sigI ≥2, Rpim ≤0.5 and CC1/2 >0.95 in their highest-resolution cell.

Isotropic data analysis:

Spacegroup	P21
Cell parameters	43.773 92.719 68.728
	90.000 90.006 90.000
Wavelength [Å]	0.97926

	Overall	Inner Shell	Outer Shell
Low resolution limit	68.728	68.728	1.585
High resolution limit	1.558	4.230	1.558
Rmerge (all I+ & I-)	0.073	0.041	1.079
Rmeas (all I+ & I-)	0.080	0.044	1.235
Rpim (all I+ & I-)	0.031	0.017	0.583
Total number of observations	433339	25178	7292
Total number unique	68559	3989	1698
Mean(I)/sd(I)	14.0	36.5	1.1
Completeness	87.8	99.9	44.1
Multiplicity	6.3	6.3	4.3
CC(1/2)	0.998	0.997	0.576

Minor Grammatical suggestions

Line 55 – consider changing to ",however, none of these contain RNA or ..."

corrected

Line 87 – check grammar on "hairpin change from an open conformation to a closed conformation .." consider changing to "hairpin changes from .."

corrected

Line 183 – check grammar for "and that the mutation W330A loss this specific recognition."

corrected

Reviewer #2 (Remarks to the Author):

The manuscript described the molecular basis of GTP binding to SARS-CoV-2 N protein using protein chemistry, mutagenesis, binding affinity, and structural biology. Overall the work is clear, conclusion supported by the data and the methods are well described. The findings are interesting as this represent the first structure of the N protein form SARS-CoV-2 binding GTP and describing its specificity over other nucleotide. This could help design some drug to block the binding and disrupting the RNA packing that is the principal function of the N protein.

Comments

Table I: the beta angle for the crystal form I of NCTD crystallised in the space group P21 should be different form 90degree, there is probably a typo here. Also it look like this data set was cut in resolution with different rules than the two other data set and could have been pushed further. Can the authors please clarify?

Thank you for the comment. This is a common concern of reviewers 1 and 2. As we answered to reviewer 1, for a monoclinic crystal $\alpha=\gamma=90^\circ$ and there is no restriction for the beta angle. It is not very likely that $\beta = 90^\circ$, but this value is not forbidden. Indeed, there are >100 crystal structures in the PDB with spacegroup $P2_1$ and $\beta=90^\circ$. We were also concerned about this crystal indexing and attempted to reprocess the data and solve the structure in orthorhombic space group without success. The spacegroup $P2_1$ was selected automatically during data processing at the beamline and the structure could be resolved without further problems.

Page 4: the authors described 2 alternate conformation of the beta hairpin, with the closed conformation forming interaction with the symmetrical in the crystal packing, would this conformation be due to crystal packing? Or is only stabilised by the crystal packing?

We understand the reviewer's concern. However, the symmetrical contact present in our structures is also observed in the crystal structures solved by other groups with different space groups and cell dimensions. Previous studies suggest that the potential of self-interactions of N-CTD in crystal structures actually exist in solution (Yang et al., Front Chem 2021). In addition, the last C-terminal portion from N-CTD domain to the C-terminus (res 364-419) mediates interdimeric contacts in solution and those constructs have a MW corresponding to a tetramer in solution (Ye et al., Protein Science 2020). Therefore, we do not consider that the interaction and the conformation are a consequence of the crystal packaging.

Figure 2. The second GTP in the crystal form II (P1) seems to show weak density for the last PO4 group. Given this how clear was the omit maps for each GTP? Are the B factor of R259 and R262 similar to the PO4-gamma of the GTP? If not why?

We thank the reviewer for these questions. Based on the reviewer's concern and also in response to Reviewer 1, we modified Supplementary figure 3a to include the omit map for the different GTP molecules. The GTP-interacting residues R259 and R262 have ~2-fold lower B factors than the nucleotide gamma phosphate. These lower values are somehow expected since both residues present additional contacts with the beta phosphate and the ribose OH groups, and thus, are likely less flexible. As we describe in the text, the nucleotide anchors to the protein through the nitrogen base and shows less contacts in the phosphate moieties, specially at the last gamma phosphate, which is mostly exposed to the solvent and is expected to have higher flexibility and thus, higher B-values. We speculate that this pocket could be a potential RNA-binding region. Since the RNA polymer lacks beta and gamma phosphate, the higher B-values for these groups is somehow expected.

The omit maps were generated by Anneal method as requested and are represented in blue color for the three GTP molecules in the following picture.

Anneal Omit map of GTP molecules ($\sigma=1$, carve 1.6)

Did the protein dimerise in solution? Or during crystallisation?

The protein is a dimer in solution, as shown below, and also reported in previous publications (Ye et al., Protein Science 2020). The theoretical MW of the monomer His-tagged N^{CTD} is = 15.5 KDa, and 31 KDa for the dimer. We show below the elution profile of the protein in a size exclusion chromatography (SEC) using a Superdex 200pg 16/60 column (GE Healthcare). The retention volume is = 83.4 ml, corresponding to an empirical MW of 39.5 KDa, a slightly increased value compared to the theoretical MW of the dimer that might be due to the presence of the purification tags in the dimer and the consequent interment of the hydrodynamic radius.

Line152-153 does the 1 degree difference in T_m significant?

Figure 3c was analyzed by comparing the nonlinear fitting curve, using the Aikake test (AIC), showing that the T_m of N and T_m of N+GTP are different with a probability >99.9% and AIC value of 221.1. In conclusion the difference is statistically significant. This analysis has been described in the main text and in the Methods section.

In addition to the SL5a does the RNA of coronaviruses rich in G?

The idea we suggest here is not that the NCTD-protein binds to G-rich regions. In many single-stranded (ss) RNA viruses, the cis-acting packaging signal that confers selectivity genome packaging usually encompasses short structured RNA repeats. These structural units, termed repetitive structural motifs (RSMs), potentially mediate capsid assembly by specific RNA-protein interactions. In a recent publication (Chen et al. 2021) they identified conserved RSMs in all coronavirus subtypes and they all have a single Guanine in a conserved position. "The identification of RSM-encompassing structural elements in all CoVs suggests that these RNA elements play fundamental roles in the life cycle of CoVs. In the recently emerged SARS-CoV-2, beta-CoV-specific RSMs are also found in its SL5, displaying two copies of 5'-gUUUCGUc-3' motifs" (cited from the above publication).

Our results suggest that the G could be specifically recognized by the C-terminal domain of the N protein. However, further studies, ideally including the X-ray structure of this protein domain in complex with one RSM, will address this specificity.

If drugs were to be developed against this GTP binding, how different is the GTP binding of the NCTD compare to other GTP binding protein (specially human one)?

We thank reviewer 2 for this question that prompted us to add some interesting data to the manuscript. We had used the PDB advanced search to identify human proteins with structure similar to N-CTD of SARS-CoV-2, and containing GTP or its non-hydrolysable analog, GNP. The guanine binding site of N-CTD was compared with the following protein structures:

Human Rho-related GTP-binding protein Rho6 in complex with GNP (PDB ID: 2REX).

Human Ras related protein Ral-A in complex with GNP (PDB ID: 1ZC3).

Human CD38 in complex with GTP (PDB ID: 3DZI).

The human structures and the N-CTD bound to GTP were superposed over the guanine base. All three human proteins showed a different orientation of the triphosphate moiety respect to the GTP molecule of N-CTD. As described in the manuscript, the guanine base in N-CTD is bound through a π - π T-shaped stacking with a tryptophan residue (W330) in one face, and the acetamide group of R259 stacks in the opposite face of the nitrogen base. At the bottom of the pocket, the guanine establishes polar contacts with the main chain of the residues of β 2 (panel

A). In contrast, all three human structures show negative or polar residues (D,E,S) located at the bottom of the pocket (panel B,C,D). In Rho6 (panel B) and Ral-A (panel C) there is a lysin residue that stacks at one face of the guanine base, but there is no aromatic residue mediating the interaction that could mimic the W330 of N-CTD. In the opposite face of the guanine, there are no arginine residues mediating the interaction, but tyrosine and phenylalanine instead. In human CD38, two tryptophan residues interact with the guanine (panel D). An inclined tryptophan (W125) in one face of the guanine, but with a different rotamer than the W330 of N-CTD, and the second tryptophan (W189) makes a π - π stacking (face-to-face) interaction with the guanine.

We did not find any other human protein structure, in complex with GTP, with higher structural similarity to N-CTD than those presented here.

We have added the following figure to the work (Supplementary figure 7) and referred to it in the discussion section of the manuscript as shown below:

Supplementary figure 7. Comparison of the GTP binding site of N^{CTD} with structurally similar human proteins. **a)** Close view of the GTP binding site of SARS-CoV-2 N^{CTD}, **b)** Human Rho-related GTP-binding protein Rho6 in complex with GNP (PDB ID: 2REX), **c)** Human Ras related protein Ral-A in complex with GNP (PDB ID: 1ZC3), **d)** Human CD38 in complex with GTP (PDB ID: 3DZI). The residues that mediate interaction with the nucleotide are represented in sticks with carbon atoms in orange. Ligands are represented in sticks, with carbon atoms colored in black. Nitrogen, oxygen and phosphorous atoms are colored in blue, red and orange, respectively.

In addition, the comparison of the guanine-binding pocket of NCTD with structurally similar human proteins that also bind guanine nucleotides, such as Rho6, Ral-A, and CD38 (Supplementary Figure 7) shows low conservation in the set of residues and their structural organization for the nucleotide binding, suggesting that the design of specific compounds against SARS-CoV-2 NCTD may be feasible.

Reviewer #3 (Remarks to the Author):

The manuscript “The identification of a guanine-specific hydrophobic pocket 1 in the N protein from SARS-CoV-2” by Ciges-Tomas, et al. reports crystal structure of the C-terminal domain of SARS-CoV-2 nucleocapsid (N) protein, N-CTD, and the protein in complex with GTP. They identify W330 as a crucial residue for GTP interaction and W330A mutant abrogated the binding of oligonucleotides. Overall, the work is of good quality. However, the language needs improvement. My specific comments are:

1. On page 6, line 122: “... and other nitrogen base would not be easily accommodated in the pocket.” The authors should show in Supplementary Figure that there will be steric hindrance when other bases (A, U or C) are put in that binding site.

We thank reviewer 3 for the suggestion. A new supplementary figure shown here (Supplementary Figure 4) has been introduced superposing ATP, UTP, and CTP over the nucleotide GTP. The models show the lacking of favorable interactions for the correct accommodation of other ribonucleotides in the protein pocket.

2. Page 6, line 123-128: “.....The structure suggests that the GTP binding favours swapping of the β -hairpin and interdimeric interaction.” What do authors mean by swapping of beta-hairpin and inter dimeric interactions? The C-terminal tail of a symmetry-related dimer coming close to GTP binding site would be a result of crystal contacts and would not enhance the binding of GTP.

According to our results and other published structures, residue W330 can swap between the open and close conformations. Within this freedom, we agree that the Pro residue of the symmetry-related C-tail could interact with W330 in a closed state. However, when GTP is bound, W330 is locked in the close conformation, interacting with the Pro residue of the symmetry-related C-ter. Thereby, the structures suggest that the GTP binding could favor the interdimeric interaction, by locking the W330 in the closed state.

3. Page 7, line 142: “Whereas GTP decreased significantly the quenching slope of NCTD (from

6.31 ± 0.08 to 5.17 ± 0.09; p-val ****)” The authors mention the p-value but number of replicates for the experiments should also mentioned either in the main text or in Supplementary information.

The experiments have been repeated at least four times in an independent manner. This is now refereed in the revised text. Statistical values of the slope comparison, including n values, are indicated in Supplementary Table III.

We included the following the text, lines 139-147:

*“We determined the quenching of the fluorescence in absence or presence of 0.5 mM GTP. Whereas GTP decreased significantly the quenching slope of N^{CTD} (from 6.31 ± 0.08 to 5.17 ± 0.09; p-val ****, n=7 and n=4), the quenching of N^{CTD-W330A} (1.99 ± 0.06; n=4) is not affected by the presence of GTP (1.74 ± 0.06; n=4), indicating that only the accessibility of W330 is affected by GTP (Figure 3b and Supplementary Tables II and III)”*

4. Page 8, line 179-182: “EMSA assays showed that the mutant binds less efficiently to the oligonucleotide Oligo-G than the NCTD-wt (Kd=130 ± 34 nM, n=3), and with a similar affinity to Oligo-U (Kd=90 ± 19 nM)”. It appears that the Kd for oligo-G binding to Nctd-wt is 130 nM, which is not true. The sentence may be reframed....

The sentence has been rephrased as follows: “EMSA assays showed that the mutant W330A binds less efficiently to the oligonucleotide Oligo-G (Kd = 130 ± 1 nM, n= 3) than the NCTD-wt to Oligo-G (Kd=32 ± 5 nM, n=3), and with a similar affinity to Oligo-U (W330A:Oligo-U Kd=90 ± 19 nM, n=3 and NCTD-wt:Oligo-U Kd = 140 ± 34 nM, n=3)”.

REVIEWERS' COMMENTS:

Reviewer #1 (Remarks to the Author):

The authors have addressed all of my concerns and I can recommend for publication.

Reviewer #2 (Remarks to the Author):

The authors have provided satisfactory answers to my questions, additional details and figures to further explain their results. I have no further query.

Reviewer #3 (Remarks to the Author):

In the revised manuscript, the authors have taken into account the concerns raised by me earlier.